# Insights into the Role of a Cardiomyopathy-Causing Genetic Variant in *ACTN2*

**DOI:** 10.3390/cells12050721

**Published:** 2023-02-24

**Authors:** Sophie Broadway-Stringer, He Jiang, Kirsty Wadmore, Charlotte Hooper, Gillian Douglas, Violetta Steeples, Amar J. Azad, Evie Singer, Jasmeet S. Reyat, Frantisek Galatik, Elisabeth Ehler, Pauline Bennett, Jacinta I. Kalisch-Smith, Duncan B. Sparrow, Benjamin Davies, Kristina Djinovic-Carugo, Mathias Gautel, Hugh Watkins, Katja Gehmlich

**Affiliations:** 1Institute of Cardiovascular Sciences, University of Birmingham, Birmingham B15 2TT, UK; 2Division of Cardiovascular Medicine, Radcliffe Department of Medicine and British Heart Foundation Centre of Research Excellence Oxford, University of Oxford, Oxford OX3 9DU, UK; 3Department of Physiology, Faculty of Science, Charles University, 12800 Prague, Czech Republic; 4Randall Centre for Cell and Molecular Biophysics, King’s College London, London SE1 9RT, UK; 5School of Cardiovascular and Metabolic Medicine and Sciences, British Heart Foundation Centre of Research Excellence, King’s College London, London SE1 9RT, UK; 6Department of Physiology, Anatomy and Genetics, University of Oxford, Oxford OX1 3PT, UK; 7Transgenic Core, Wellcome Centre for Human Genetics, University of Oxford, Oxford OX3 7BN, UK; 8European Molecular Biology Laboratory, 38000 Grenoble, France; 9Department of Structural and Computational Biology, Max Perutz Labs, University of Vienna, 1030 Vienna, Austria; 10School of Basic and Medical Biosciences, British Heart Foundation Centre of Research Excellence, King’s College London, London SE1 9RT, UK

**Keywords:** alpha-actinin, embryonic heart, sarcomere, cardiomyopathy, proteomics, mitochondria

## Abstract

Pathogenic variants in *ACTN2*, coding for alpha-actinin 2, are known to be rare causes of Hypertrophic Cardiomyopathy. However, little is known about the underlying disease mechanisms. Adult heterozygous mice carrying the *Actn2* p.Met228Thr variant were phenotyped by echocardiography. For homozygous mice, viable E15.5 embryonic hearts were analysed by High Resolution Episcopic Microscopy and wholemount staining, complemented by unbiased proteomics, qPCR and Western blotting. Heterozygous *Actn2* p.Met228Thr mice have no overt phenotype. Only mature males show molecular parameters indicative of cardiomyopathy. By contrast, the variant is embryonically lethal in the homozygous setting and E15.5 hearts show multiple morphological abnormalities. Molecular analyses, including unbiased proteomics, identified quantitative abnormalities in sarcomeric parameters, cell-cycle defects and mitochondrial dysfunction. The mutant alpha-actinin protein is found to be destabilised, associated with increased activity of the ubiquitin-proteasomal system. This missense variant in alpha-actinin renders the protein less stable. In response, the ubiquitin-proteasomal system is activated; a mechanism that has been implicated in cardiomyopathies previously. In parallel, a lack of functional alpha-actinin is thought to cause energetic defects through mitochondrial dysfunction. This seems, together with cell-cycle defects, the likely cause of the death of the embryos. The defects also have wide-ranging morphological consequences.

## 1. Introduction

Contractility in heart and skeletal muscle is achieved through highly organised structures, the sarcomeres, in which actin-rich thin filaments and myosin-rich thick filaments slide against each other. Both filaments have specially organised anchoring sites: the Z-disk for thin filaments, and the M-band for thick filaments [1,2]. In addition, single titin molecules span from Z-disks to the M-band as a third filament [3].

The Z-disk is a highly organised and complex structure. In it, alpha-actinin cross-links antiparallel actin filaments from adjacent sarcomeres and organises them, together with other actin cross-linking proteins, e.g., filamin C, into a highly ordered ultrastructural paracrystalline structure. Beyond its mechanical role, the Z-disk is recognised as a signaling hub, which integrates mechano-signaling and stress responses. Numerous proteins with signaling roles are known to be residents of the Z-disk, sometimes in a transient fashion [4].

The highly conserved alpha-actinin is a key protein of the Z-disk. It features an amino-terminal actin-binding domain consisting of two calponin homology domains (CH1, CH2) and a carboxy-terminal calmodulin-like domain. The middle rod domain, comprised of four spectrin-like repeats, allows it to form anti-parallel dimers. Titin binding is regulated via phospho-lipids (PIP2) to the actin-binding domain, triggering a conformational change of the calmodulin-like domain, leading to an open conformation capable of binding to titin Z-repeats [5]. The interaction of alpha-actinin 2 with actin filaments was shown to require the opening of the actin binding domain, where only the CH1 domain binds to the filament, while the second is dissociated from the interacting domain and acts as a negative regulator of the interaction [6,7]. This structural mechanism is the basis of the explanation of many mutations that map to the interface between CH1 and CH2 domains.

There are four genes coding for alpha-actinin (*ACTN1–ACTN4*). *ACTN2* is highly expressed in heart and skeletal muscle, and *ACTN3* in some skeletal muscle subtypes. *ACTN1* and *ACTN4* are found predominantly in non-muscle cells, where they perform similar actin cross-linking functions. In line with their expression pattern, *ACTN2* genetic variants have been described to cause autosomal dominant skeletal muscle and cardiac diseases [2]. Four missense variants in *ACTN2* (p.Gly111Val, p.Ala119Thr, p.Met228Thr and p.Thr247Met) have been identified in the actin-binding domain in patients with Hypertrophic Cardiomyopathy (HCM) [8,9,10,11], a heart muscle disease that results in stiffer and hypertrophied hearts and can be associated with life-threating arrhythmias. Of note, both *ACTN2* p.Met228Thr and p.Thr247Met map to the interface between CH1 and CH2 of the actin binding domain in the closed conformation. For *ACTN2* p.Ala119Thr, p.Met228Thr and p.Thr247Met, there is genetic evidence of pathogenicity through co-segregation in a multi-generation family [8,9,10], while the pathogenicity of *ACTN2* p.Gly111Val is only supported by functional biochemical studies [12]. Moreover, cellular studies using patient-derived induced pluripotent stem cell-derived cardiomyocytes support the role for the pathogenicity of *ACTN2* p.Thr247Met [10,13].

No animal models have been generated to study these HCM-causing *ACTN2* variants in vivo. Here, we present the generation of an *Actn2* mouse model harboring the p.Met228Thr variant. In the heterozygous setting, mature male mice show molecular features in keeping with HCM. To our surprise, the homozygous *Actn2* p.Met228Thr mice were found to have an embryonic lethal phenotype. A detailed analysis of the embryonic hearts at E15.5 suggests that alpha-actinin 2 not only controls heart morphology during development but also affects cell cycling and mitochondrial function. Moreover, we could identify mutant *ACTN2* protein instability as a driving factor of the phenotype.

## 2. Materials and Methods

### 2.1. Ethical Statement

The animal studies have been performed in accordance with the ethical standards laid down in the 1964 Declaration of Helsinki and its later amendments. Experimental procedures were performed in accordance with the Directive 2010/63/EU and UK Home office guidelines (project licences P572C7345 and PDCE16CB0) and approved by the respective institutional ethical review boards.

Animals were housed in specific pathogen-free conditions, with the only reported positives on health screening over the entire time course of these studies being for *Tritrichomonas* sp. and *Entamoeba* spp. All animals were housed in social groups of mixed genotypes, provided with food and water ad libitum, and maintained on a 12 h light:12 h dark cycle (150–200 lux cool white LED light, measured at the cage floor).

Phenotyping experiments and offline analysis were performed blinded. All in vivo phenotyping studies of adult mice were carried out using littermates and both sexes. Animals were sacrificed by cervical dislocation and death was confirmed by the cessation of circulation.

### 2.2. Generation of the Mice and Genotyping

The p.Met228Thr variant was introduced into the orthologous position in the mouse *Actn2* gene using CRISPR-Cas9-mediated homology-directed repair in mouse embryonic stem cells. A detailed methodology is provided in the Appendix A.

Heterozygous *Actn2* p.Met228Thr mice were viable and fertile. The expression of the mutated allele p.Met228Thr was confirmed at the protein level by mass spectrometry (Appendix A). Animals were backcrossed onto C57BL/6J (Envigo, London, UK) for at least six generations before generating wild-type and heterozygous littermates, or embryos; aged animals were backcrossed for at least two generations.

Animals were genotyped for Actn2 p.Met228Thr mutation and a spontaneous genetic variant in the Nnt gene, occurring in C57BL/6J sub-strains [14] using Transnetyx services (Cordova, TN, USA). All animals of the colony were homozygous for the genetic variant in Nnt.

### 2.3. Ultrasound Echocardiography

Ultrasound echocardiography was carried out as previously described [15]. A detailed methodology is provided in the Appendix A.

### 2.4. Embryo Collection and Fixation, Theiler Staging

Mouse embryos were collected at E15.5 following timed mating. Once embryos were drained of blood, hearts were either dissected and flash frozen in liquid nitrogen or used for High Resolution Episcopic Microscopy (HREM) and wholemount staining. For Theiler staging, PFA-fixed front limbs were examined under an inverted stereoscope (Tl3000 Ergo stereoscope, Leica, Mannheim, Germany) and classified according to [16]. A detailed methodology is provided in the Appendix A.

### 2.5. Proteasomal Activity Assays

Chymotrypsin-like, Caspase-like and Trypsin-like activity assays were performed as previously described using commercially available indirect enzyme-based luminescent assay kits (Promega, Madison, WI, USA) [17]. A detailed methodology is provided in the Appendix A.

### 2.6. High Resolution Episcopic Microscopy

Tissue and data processing for HREM was performed as previously described [18]. A detailed methodology is provided in the Appendix A.

### 2.7. Immunofluorescence Staining on Cryosections

The staining of the cryosections of skeletal muscle was performed as described [5]. Immunolabelling with phospho-histone H3 (Thermo Fisher Scientific, Waltham, MA, USA), alpha-actinin (Sigma-Aldrich, St. Louis, MO, USA) and DAPI was carried out on cryosections of E15.5 hearts. A detailed methodology is provided in the Appendix A.

### 2.8. Ex Vivo Studies

Tibial length measurements, mRNA isolation from ventricular tissue, reverse transcriptase and quantitative PCR (qPCR) were performed as described [19], using the Taqman probes (Applied Biosystems, Waltham, MA, USA) listed in Appendix A. Western blotting was performed as described [19] with the antibodies listed in Appendix A.

Histology on paraffin-embedded samples (7 µm sections) was performed with hematoxylin and eosin, using standard protocols.

### 2.9. Mass Spectrometry for Identification

An excised gel sample was digested with trypsin (Promega, Madison, WI, USA) and analysed by nano-UPLC–MS/MS using a Dionex Ultimate 3000 coupled online to an Orbitrap Fusion Lumos mass spectrometer (Thermo Fisher Scientific, Waltham, MA, USA), as described [20]. A detailed methodology is provided in the Appendix A.

### 2.10. Proteomics

Wildtype and homozygous (*n* = 6 per group) ventricular samples, previously collected from embryos at E15.5, were prepared and labelled using solutions provided in EasyPepTM mini MS sample prep kit (Thermo Fisher Scientific, Waltham, MA, USA) following manufacturer’s instructions. For data analysis, the MS and MS/MS scans were searched against the Uniprot database using Proteome Discoverer 2.2 (Thermo Fisher Scientific, Waltham, MA, USA) with 5% false discovery rate (FDR) criteria. Multiple test correction was performed using the Benjamini and Hochberg test (Perseus software [21]). Data were further analysed through the use of Ingenuity pathway analysis (IPA, QIAGEN Inc, Germantown, MD, USA), https://digitalinsights.qiagen.com/IPA, (accessed on 22 February 2023) [22]). Detailed information is provided in the Appendix A.

The proteomics data underlying this article are available in PRIDE [https://www.ebi.ac.uk/pride/archive/, (accessed on 3 January 2023)], and can be accessed with identifier PXD039226.

### 2.11. Wholemount Immunohistochemistry

The hearts were dissected from PFA-fixed embryos in PBS and treated with hyaluronidase (1 mg/mL in PBS, Sigma-Aldrich, St. Louis, MO, USA) for 45 min at RT. After three washes in PBS, permeabilization with 0.2% Triton X-100 was performed for 45 min. Following another three washes in PBS and blocking for 30 min with 5% pre-immune goat serum (Sigma-Aldrich, St. Louis, MO, USA) in antibody dilution buffer (10 mM Tris-HCL, pH 7.2, 155 mM NaCl, 2mM EGTA, 2 mM MgCl_2_, 1% BSA), the hearts were incubated with the primary antibody mixture (see Appendix A) overnight at 4 °C. After 5 × 20 min of washing in PBT (PBS with 0.002% Triton X-100), the secondary antibodies (Cy3-goat anti mouse Igs, multilabelling quality; Cy2-goat anti rabbit Igs, multi-labelling quality, both Jackson Immunochemicals, West Grove, PA, USA via Stratech Scientific, Ely, UK, DAPI, Sigma-Aldrich, St. Louis, MO, USA and Alexa647-phalloidin, Thermo Fisher Scientific, Waltham, MA, USA) were applied overnight at 4 °C. The hearts were washed in PBT for 5 × 20 min and mounted in 0.1 M Tris–HCl (pH 9.5) and glycerol (3:7), with 50 mg/mL of N-propyl-gallate as an anti-fading agent. Microscopy was carried out using an SP5 confocal (Leica, Mannheim, Germany), equipped with 405 blue diode, argon and helium-neon lasers, using a 63/1.4NA oil immersion lens.

The hearts were dissected from PFA-fixed embryos in PBS and treated with hyaluronidase (1 mg/mL in PBS, Sigma-Aldrich, St. Louis, MO, USA) for 45 min at RT. After three washes in PBS, permeabilization with 0.2% Triton X-100 was performed for 45 min. Following another three washes in PBS and blocking for 30 min with 5% pre-immune goat serum (Sigma-Aldrich, St. Louis, MO, USA) in antibody dilution buffer (10 mM Tris-HCL, pH 7.2, 155 mM NaCl, 2mM EGTA, 2 mM MgCl_2_, 1% BSA), the hearts were incubated with the primary antibody mixture (see Appendix A) overnight at 4 °C. After 5 × 20 min of washing in PBT (PBS with 0.002% Triton X-100), the secondary antibodies (Cy3-goat anti mouse Igs, multilabelling quality; Cy2-goat anti rabbit Igs, multi-labelling quality, both Jackson Immunochemicals, West Grove, PA, USA via Stratech Scientific, Ely, UK, DAPI, Sigma-Aldrich, St. Louis, MO, USA and Alexa647-phalloidin, Thermo Fisher Scientific, Waltham, MA, USA) were applied overnight at 4 °C. The hearts were washed in PBT for 5 × 20 min and mounted in 0.1 M Tris–HCl (pH 9.5) and glycerol (3:7), with 50 mg/mL of N-propyl-gallate as an anti-fading agent. Microscopy was carried out using an SP5 confocal (Leica, Mannheim, Germany), equipped with 405 blue diode, argon and helium-neon lasers, using a 63/1.4NA oil immersion lens.

### 2.12. Electron Microscopy

Embryos were generated as described above. The dissection of whole hearts was carried out in fresh ice-cold PBS with 5mM EDTA. Once cleaned of surrounding tissues, hearts were briefly flushed with ice-cold PBS via the aorta. Hearts were fixed in 4% PFA for 15 min before being transferred to 2.5% glutaraldehyde/2% PFA for 2 h at room temperature, then 3 h at 4 °C. Subsequently, hearts were stored in 0.05% glutaraldehyde at 4 °C.

Fixed hearts were briefly washed with fresh PBS and further dissected to remove the atria and blood vessels leaving the ventricles. This portion was cut crossways to produce a small tip fragment and a figure of eight fraction with most of the ventricular walls. These were further fixed in 1% osmium, dehydrated in ethanol, and embedded in Araldite. Before embedding, the larger fragment was divided into three parts: the left and right ventricular walls and the septum. Then, 70 nm sections were stained with Uranyless heavy metal stain followed by Pb Citrate (both Labtech International Ltd., Heathfield, UK). The sections were viewed in a JEOL 1400 electron microscope in the Centre for Ultrastructural Imaging, KCL.

### 2.13. Image Analysis

ImageJ version 1.53a was used for the colocalisation analysis and densitometry of Western blots. The ‘JACoP’ plugin (https://imagej.net/plugins/jacop, accessed on 22 February 2023) was used to calculate a correlation coefficient. The Image J plugin ‘Colocalisation finder’ (http://punias.free.fr/ImageJ/colocalization-finder.html, accessed on 22 February 2023) was used to generate cytofluorograms to visualise colocalisation.

Cell Profiler 4.2.1. was used for nuclear assessment [23]. Sarcomere length was analysed as described [24], using MatLab (version 2021a) and the script ‘ZLineDetection’ (https://github.com/Cardiovascular-Modeling-Laboratory/zlineDetection, accessed on 22 February 2023).

### 2.14. Statistics

All values are given as mean ± standard error of mean (SEM). To compare two unpaired sample groups, data were tested for normality using the Kolmogorov–Smirnov test. Normally distributed data were analysed by Student’s *t*-test, and data that were not were analysed by Mann–Whitney U-test. Deviation from the expected Mendelian ratios and Theiler stages were assessed with the Chi-square test. For the comparison of the three groups with normally distributed data, a one-way ANOVA followed by Tukey’s post-hoc test was used. For the image analysis of wholemount staining, nested ANOVA was employed, allowing us to consider the number of measurements from each heart. Fisher’s exact test was used to test for the occurrence of VSD. All statistical analyses were performed with GraphPad Prism 9.3.1.

Annotations used: * *p* < 0.05, ** *p* < 0.01, *** *p* < 0.001, **** *p* < 0.0001 versus WT, otherwise considered not significant (*p* > 0.05); *n* indicates number of animals in each group.

## 3. Results

### 3.1. Generation of a Mouse Model for a Hypertrophic Cardiomyopathy-Causing Genetic Variant

In order to investigate the disease mechanisms causing HCM, the p.Met228Thr substitution was introduced into the *Actn2* gene in mice by CRISPR/Cas9 genome-editing; successful engineering was confirmed by Sanger sequencing, and backcrossing eliminated CRISPR/Cas9 off-target effects. Heterozygous mice were viable and fertile. The alpha-actinin 2 protein carrying the missense change was detected in the heart by mass spectrometry (Appendix A), indicating the successful generation of the model.

### 3.2. Cardiac Phenotyping of Adult Mice with the Actn2 p.Met228Thr Variant

Young adult mice (3 months) underwent cardiac phenotyping by echocardiography. Mice carrying the heterozygous *Actn2* p.Met228Thr variant had normal cardiac function and dimensions when compared to their wildtype (WT) littermates, irrespective of their sex (Appendix A). In agreement, these mice had normal heart weights when normalised to tibial length (Appendix A). At the molecular level, they showed no induction of the fetal gene programme typically seen in cardiomyopathic hearts (Appendix A). The histology on these mice was unremarkable (Appendix A).

In more mature mice (>34 weeks), cardiac function and dimensions were again normal in both sexes when compared to their WT littermates (Appendix A). In support, mice had normal heart weights when normalised to tibial length (Appendix A). However, male mice showed a significant induction of *Nppb* and *Acta1*, two transcripts of the fetal gene programme (Figure 1, determined at 38 weeks), while female mice did not (Appendix A). Moreover, transcripts associated with hypertrophic signaling were induced in male hearts (Figure 1, significant for *Fhl1, Ankrd1* and *Ankrd2*), but not in female hearts (Appendix A, apart from a less than two-fold increase in *Ankrd2*). At the protein level, mature male mice displayed an increased expression of small heat shock protein HspB7 (Appendix A), but not of Hsp27, and a trend of increased beta-myosin heavy chain expression.

The skeletal muscle of the mice did not show any signs of pathology, regardless of genotype (Appendix A).

In conclusion, the *Actn2* p.Met228Thr variant does not produce an overt cardiomyopathy phenotype in mice, however, mature male mice display molecular features consistent with HCM.

### 3.3. Actn2 p.Met228Thr Is Embryonic Lethal in the Homozygous Setting

We next attempted to generate mice homozygous (Hom) for the *Actn2* p.Met228Thr change by crossing heterozygous mice. Nine breeding pairs produced 18 litters with 83 weaned pups, comprising of 24 WT and 59 heterozygous animals, however, no Hom *Actn2* p.Met228Thr offspring were found (Appendix A, Appendix A). This was a clear deviation from the expected Mendelian ratios (Chi-square test, *p* < 0.0001). Four further litters from timed matings were collected at the time of birth and again failed to identify Hom offspring (Appendix A, Appendix A), suggesting that Hom *Actn2* p.Met228Thr are embryonic lethal.

As the next step, we performed timed matings to harvest embryos at defined time points; E15.5 was the latest time point that viable Hom embryos were consistently present. At this timepoint, 30 litters produced 242 embryos, among them 56 Hom (Appendix A, Appendix A). This was within the expected Mendelian ratios (Chi-square test, *p* > 0.05).

### 3.4. Detailed Morphological Analysis of Hom Actn2 p.Met228Thr E15.5 Embryos

The gross morphology and size of E15.5 embryos appeared normal regardless of the genotype (Appendix A).

HREM was performed on E15.5 wildtype and Hom hearts. Strikingly, Hom hearts had significantly increased right ventricle luminal volume (*p* = 0.0006) and smaller left atrium volume (*p* = 0.0030) when compared to control hearts (Figure 2B and Appendix A). In addition, Hom hearts had significantly reduced left ventricular free wall thickness (*p* = 0.0021, Appendix A), with a visual trend of reduced wall thickness in two other areas of LV and RV (Appendix A), although this did not reach statistical significance. Three out of eight Hom hearts had peri-membranous ventricular septal defects (Figure 2A). Despite the occurrence only in Hom hearts, this was not statistically significant (*p* = 0.10, Fisher’s exact test). No atrial septal defects were observed. Although the aortic arch and pulmonary trunk of Hom embryos had normal gross morphology (Appendix A), the pulmonary trunk volume was significantly reduced (*p* = 0.0499, Appendix A), despite normal length (*p* = 0.52, Appendix A). Furthermore, significant aortic stenosis was evident in both the ascending

(*p* = 0.0025, 19%) and descending thoracic aorta (*p* = 0.0497, 32%, Appendix A), but not in the aortic root (*p* = 0.0609, Appendix A). Finally, dysplastic pulmonary valves with thickened leaflets were observed, particularly in the right and left leaflets (Appendix A).

Hom embryos were found to have malpositioned hearts within the body cavity, with a superior tilt of the heart (*p* = 0.0104, Appendix A–C). Affected hearts also tended to have a leftward rotation (when measured from the base of the aorta to the ventricular sulcus), but this was not statistically significant.

In summary, we identified significant morphological changes in the ventricular chambers, aorta, pulmonary valve and pulmonary trunk in the hearts of Hom *Actn2* p.Met228Thr embryos, with a subset of embryos also having a peri-membranous ventricular septal defect.

To exclude developmental delay as an explanation for the observed abnormalities in the Hom embryos, forelimb morphology was used for Theiler staging, and no developmental delay was observed (Appendix A, Chi-square test, *p* > 0.05).

### 3.5. Sarcomeric and Nuclear Organisation in Hom Hearts

Wholemount immunofluorescence was performed to interrogate the expression and localisation of alpha-actinin 2 in E15.5 hearts. Of note, residual congealed blood was observed macroscopically in the ventricles of Hom hearts, much more than in the WT hearts (Appendix A), suggesting the less efficient contractility of Hom hearts during the fixation procedure.

Sarcomeric structures were visualised with titin Z-disk epitope (T12) and were clearly present and well organised, with alpha-actinin detectable at the Z-disks of sarcomeres in both genotypes (Figure 3A). However, quantitative colocalisation analysis revealed reduced colocalisation of titin Z-disk epitope with alpha-actinin (Figure 3B,C). Moreover, sarcomere length was found to be substantially increased in the Hom hearts (2.2 versus 1.8 µm; Figure 4A). In addition, there was a reduction in the number of nuclei in the Hom hearts, and they were rounder in shape (Figure 4B, Appendix A).

Electron microscopy demonstrated regular Z-disks in both WT and Hom samples (Appendix A), however, Z-disks appeared less uniform in the Hom hearts (Appendix A). While Z-disk measurements revealed no difference in Z-disk width between both genotypes (*p* = 0.12), the width distribution was much wider in the Hom hearts (Appendix A).

### 3.6. Cell-Cycle Defects in Hom Hearts

The reduced number of nuclei (Figure 4B) prompted us to investigate cell-cycle markers. In a targeted transcript analysis for cell-cycle markers, *Anln, Cdkn1a, Cdkn1b, Cdkn2b, Tp53* and *Wee1* were all found to be dysregulated in the ventricles of Hom hearts (Figure 4C), with the most striking upregulations being observed for *Anln, Cdkn1a* and *Wee1*, which were all predicted to block the progression of the cell cycle. In support of *Tp53* transcript upregulation, a non-significant trend (*p* = 0.13) towards the upregulation of the p53 protein level was observed on Western blotting (Appendix A).

To further probe for potential defects in cell division in the Hom hearts, we performed staining for phosphorylated histone H3, a nuclear marker of active proliferation in cells (Figure 5A). In Hom E15.5 hearts, fewer cells positive for the marker were observed (Figure 5B). An analysis of dividing cells at higher magnification identified clear evidence of metaphase chromosome arrangement in dividing Hom embryonic cells, but these cells seemed to have a defect in myofibril disassembly, with residual sarcomeres observed (Appendix A).

### 3.7. Proteomics Analysis Gives Insights into Disturbances in the Hom Hearts

In order to gain insight into the disturbances leading to embryonic death, WT and Hom E15.5 heart protein samples were subjected to unbiased proteomics. Overall, 244 proteins were found to be upregulated and 133 proteins were downregulated in the Hom hearts when compared to WT (Figure 6A, Appendix A). Further, ingenuity pathway analysis (IPA) revealed dysregulation in a number of key canonical pathways, the most prominent being oxidative phosphorylation, mitochondrial dysfunction, sirtuin signalling and the citric acid (TCA) cycle (Figure 6B). Moreover, proteins belonging to the ubiquitination pathway and unfolded protein response were also enriched in the dataset (Figure 6B, Appendix A). A deeper analysis of the data highlighted key mitochondrial protein complexes to be downregulated (Figure 6C and Appendix A), while proteins associated with protein processing and translation, including proteasomal activity, were upregulated (Figure 6B,C).

### 3.8. Destabilisation of Actn2 Protein in the Hom Hearts

A targeted interrogation of the proteomics dataset indicated that Actn2 protein levels were reduced by approximately 25% in the Hom hearts (*p* < 0.01, Appendix A). To validate this, we probed for alpha-actinins 1–4 at the mRNA level. The transcripts for *Actn2* and *Actn3* were upregulated in Hom ventricles (Figure 7A and Appendix A), while transcripts for other alpha-actinins, *Actn1* and *Actn4*, were not affected. Equally, the expression of cardiac actin, *Actc1*, was normal, while skeletal muscle actin, *Acta1,* was downregulated.

In contrast to the *Actn2* upregulation observed at the transcript level, the protein levels for Actn2 were significantly reduced (Figure 7B). At the protein level, Actn3 expression was unchanged (Appendix A).

The reduced Actn2 levels in the Hom hearts suggested that the p.Met228Thr Actn2 is subject to protein degradation. Ubiquitin-proteasomal system (UPS) and autophagy are the main protein degrading machineries in cells [25]. We probed for autophagy makers (p62 and LC3, Appendix A), but failed to observe any changes in the Hom hearts. Furthermore, proteolytic activities and total ubiquitination showed no signs of changes in the Hom hearts (Appendix A). However, the fact that the UPS was enriched in the proteomics dataset (Figure 6C) argues for the UPS being responsible for the destabilisation of the Actn2 protein.

In summary, the p.Met228Thr genetic variant renders the protein less stable. In homozygous animals, cell-cycle defects and mitochondrial dysfunction are observed; these impaired functions result in a range of morphological abnormalities of the embryonic hearts and are incompatible with life.

## 4. Discussion

The heterozygous *ACTN2* p.Met228Thr variant was originally identified in a four-generation Italian family with atypical Hypertrophic Cardiomyopathy [9]. The *ACTN2* variant showed co-segregation with cardiomyopathy, consistent with autosomal dominant inheritance, across 11 affected and 7 healthy family members, and was hence considered pathogenic. A detailed clinical investigation of affected family members revealed a range of cardiac features, including left ventricular hypertrophy, restrictive pathology, arrhythmias (including early-onset atrial fibrillation) and non-compaction.

In order to gain insights into the disease mechanisms of the *ACTN2* p.Met228Thr variant, we generated a mouse model. Young adult mice heterozygous for the variant—reflecting the genetic situation in the patients—had no cardiac phenotype. Despite normal cardiac dimensions and function on echocardiography, mature male mice displayed molecular transcript signatures compatible with early signs of cardiomyopathy [26]. These findings mirror previously studied mouse models of cardiomyopathy: often the genetic equivalent of the human disease is not sufficient to cause detectable phenotypes in mice [15,27,28]. Ageing can unmask disease features [15,29], which is in agreement with HCM being a late-onset disease in humans, with patients often presenting only as adults. For example, the index patient of the *ACTN2* p.Met228Thr family was diagnosed with HCM in his 50s [9], which corresponds to approximately 10 months of age in mice. We can only speculate whether further ageing or stressors, such as adrenergic stimulation or a high-fat diet, might provoke a phenotype, as shown for other animal models [30,31,32].

Sex differences in cardiac phenotypes of C57Bl6 mice are well documented [33,34]. Of note, male mice have higher mean arterial pressure, and this sex difference increases between 3 and 12 months [35]. This could contribute to the observed molecular signs of cardiomyopathy in male, but not female, mature mice.

The lack of an overt HCM phenotype in the heterozygous mouse model, equivalent to the human *ACTN2* p.Met228Thr genetic situation, might be explained by the wildtype form dominating the alpha-actinin 2 protein. In support, we identified far more wildtype peptides in mass spectrometry than peptides carrying the p.Met228Thr variant; however, we were not able to detect the ubiquitination of alpha-actinin 2 by Western blotting (Appendix A).

Given the lack of an overt phenotype in heterozygous mice, the embryonic lethality of the variant in the homozygous setting was unexpected. Our detailed morphological analysis using HREM identified a broad range of abnormalities: The Hom E15.5 hearts had enlarged right ventricles, smaller left atria and regional decreases in the left ventricle. Three of the eight homozygous hearts had peri-membranous ventricular septal defects (VSD). Further, we observed aortic stenosis, a reduced volume of the pulmonary trunk and a thickening of the pulmonary valve leaflets, as well as an abnormal orientation of the hearts in the body cavity. While some of these defects (e.g., thinner wall, larger right ventricular lumen and VSD) might be attributable to reduced cell division and/or cell migration, others may be consequences of altered hemodynamics in the developing heart: alpha-actinin 2 has no noticeable expression in the aorta or pulmonary trunk, so changes in these structures are likely to be secondary to altered blood flow. It has been demonstrated for chicken embryos and zebrafish aortas that blood flow, or rather the lack of it, can induce remodeling [36,37]. However, the presence of a VSD did not affect other parameters measured, and there was no statistical difference between Hom hearts with and without VSD.

Of note, the combination of morphological features resembles aspects of Noonan syndrome, in which a dysplastic pulmonary valve and Hypertrophic Cardiomyopathy are observed [38,39]. However, we observed no changes in ERK phosphorylation in the embryonic hearts. Hence, if there was a joint defect with Noonan’s syndrome, it is downstream of this point. Whether such a common disease pathway exists will be the subject of future investigations.

Despite the indirect evidence of poor contractility in the Hom hearts (residual blood found in ventricular cavities after fixation), the sarcomeres appeared to be well formed and qualitatively normal in the hearts. Quantitative image analysis, however, revealed the reduced colocalisation of the titin Z-disk epitope with mutant alpha-actinin. Moreover, sarcomere length was substantially increased in the Hom hearts.

At the developmental stage investigated (E15.5), cardiomyocytes undergo cycles of cell division [40], which requires them to disassemble and later reassemble sarcomeres [41]. Based on structural analysis, we hypothesise that *ACTN2* p.Met228Thr, which maps to the interface between the CH1 and CH2 domains of the actin binding domain, might have an increased affinity to F-actin and quantitatively interfere with the breakdown of the sarcomeres required to undergo cell division. In support of this hypothesis, we identified defects in myofibril disassembly in cells undergoing division (Appendix A). As a consequence, the cells in the Hom hearts appear to have a shift towards fewer nuclei, and fewer proliferating cells were identified. In addition, the cell-cycle markers *Anln, Cdkn1a* and *Wee1* were found to be upregulated in Hom hearts. All three control various cycle check points and their upregulation impairs the progression of cell division. This impairment of cell division in Hom hearts may also explain the reduced wall thickness observed in the LV wall and the lack of muscle growth, resulting in VSD in a proportion of Hom hearts.

A striking feature of the *Actn2* p.Met228Thr variant is protein destabilisation *in vivo*. At the protein level, alpha-actinin 2 was found to be reduced to approximately one-third in Hom E15.5 hearts. It appears that the Hom hearts try to compensate for this lack of protein by upregulating *Actn2* at the transcript level. Nevertheless, this cannot make up for the decreased stability at the protein level and subsequent degradation.

The mutation may lead to the partial mis-folding of the actin binding domain, which is supported by the enrichment of the ‘unfolded protein response’ pathway in the proteomics dataset (Figure 6B). Such mis-folded proteins are recognised and targeted for protein degradation. There are two major protein degrading pathways: the ubiquitin-proteasomal system (UPS) and autophagy [42]. We have no evidence to suggest that autophagy is involved. In contrast, unbiased proteomics revealed that UPS activity is more active in Hom hearts (Figure 6B,C and Appendix A), so it is likely responsible for the increased turnover of the mutant protein. While the measurements of proteolytic activities (Appendix A) do not support the increased activity of the UPS, future experiments using proteasomal or autophagy inhibitors on cultured cells, e.g., induced pluripotent stem cell-derived cardiomyocytes, will shed light on the specific role of the protein degrading pathways.

The UPS has been implicated in other HCM-causing genetic variants, e.g., for *MYBPC3* truncating variants and for *CSRP3* C58G [27,32,43]. In support of a crucial role of protein stability in the pathogenesis of human HCM due to the p.Met228Thr variant, there are clear parallels to another HCM-associated *ACTN2* variant (p.Thr247Met) located in the same domain [13]: Using an induced pluripotent stem cell-derived cardiomyocyte model, the authors observed a reduced stability of the mutant *ACTN2* p.Thr247Met protein, with the activation of both UPS and autophagy. However, while their mutant protein formed aggregates in their cellular model upon exogenous expression, we did not detect any alpha-actinin 2 aggregates in the p.Met228Thr Hom hearts.

It is worth noting that the tightly controlled UPS function is crucial for cell homeostasis and regulating many cellular processes, e.g., normal proteasomal activity is required for sarcomere disassembly during cell division in embryonic rat cardiomyocytes. If the proteasome is inhibited, alpha-actinin 2 fails to disassemble from sarcomeric structures [41].

Unbiased proteomics identified oxidative phosphorylation and mitochondrial dysfunction as the main consequences in the Hom hearts. During embryonic development at E11.5, glycolysis is the main source of energy while oxidative phosphorylation complexes begin to form and electron transport chain activity begins [44]. By E13.5, mitochondrial structure and function resemble those of mature mitochondria, and ATP is generated mainly through oxidative phosphorylation, with glycolysis becoming a secondary source [45,46]. Guo et al. [47] recently showed through the cardiomyocyte-specific mosaic expression of a hypomorphic mutation that *Actn2* expression is important for cardiomyocyte maturation via serum response factor signaling. As one of the downstream pathways, mitochondrial expansion and organisation were found to be impaired. If dysfunctional mitochondria cannot facilitate maturation and metabolic shift in our model, energy deficiency might be a cause of embryonic lethality.

Moreover, alpha-actinin 2 has been reported to more directly influence mitochondrial function. It controls the localization of the RNA transcripts required for oxidative phosphorylation via its interaction with the RNA binding protein IGF2BP2 [48].

However, an alternative explanation would be that mitochondrial defects occur secondary to cell death [49], however, this is less likely as mitochondrial impairment was also observed in a cellular model of the *ACTN2* p.Thr247Met variant [13] in the absence of pronounced cell death.

## 5. Conclusions

In summary, our heterozygous mouse model (in mature male mice) supports a pathological role of the *ACTN2* p.Met228Thr variant but provides little insight into the disease mechanisms. Differences in physiology between humans and mice, the relatively young age of the mice and lack of stressors might explain this lack of an overt phenotype [50].

However, the detailed investigations of the Hom *Actn2* p.Met228Thr embryonic hearts identified alpha-actinin protein destabilisation as a key feature of the model. This has several implications: firstly, it leads to an aberrant activity of the UPS, which has been implicated in multiple studies of HCM. Secondly, the lack of functional alpha-actinin has been suggested to interfere with serum response factor signaling [47], preventing cardiomyocyte maturation and consequently ensuing mitochondrial dysfunction. Together with the observed cell division defects, energetic deficiency is the likely cause of myocardial dysfunction. Moreover, the experiments identified a range of morphological abnormalities in the developing heart, suggesting that subtle functional abnormalities in alpha-actinin can have major consequences on the structure and function of sarcomeres as well as cell division, and consequently on the developing heart.

## Figures and Tables

**Figure 1 cells-12-00721-f001:**
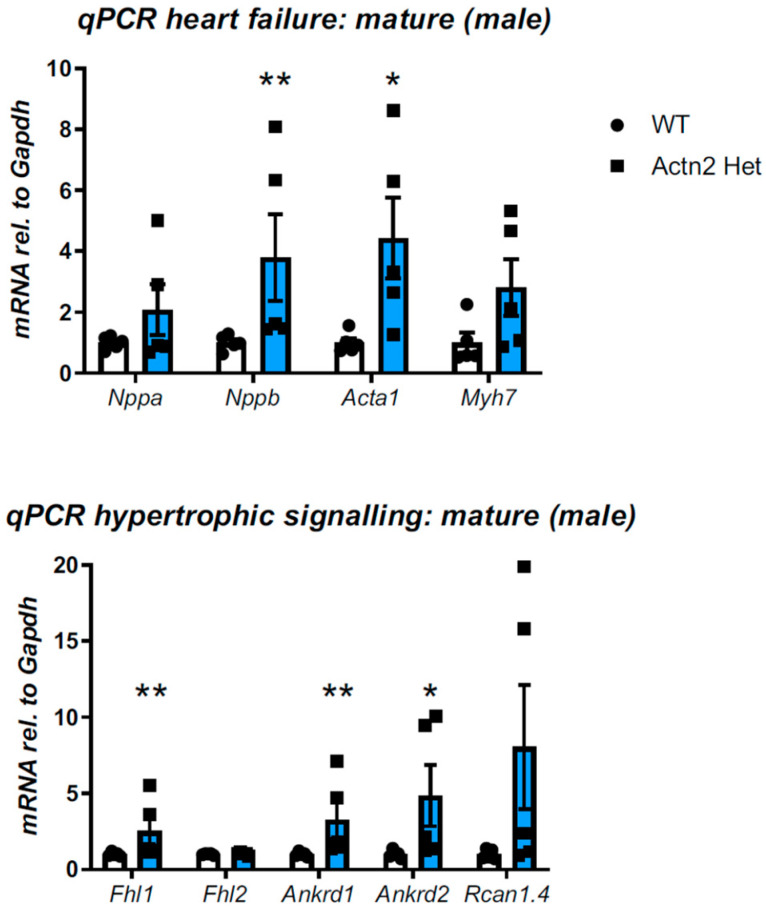
Targeted assessment of transcriptional changes by qPCR in mature male Het and WT hearts. Top: a panel of transcripts related to heart failure (fetal gene programme), Bottom: a panel related to hypertrophic signaling. All measurements are normalised to *Gapdh*. Significant changes are observed in the hearts of Het mice (Mann–Whitney U-test, *n* = 5 per group, age 266+/−1 d for both genotypes * *p* < 0.05, ** *p* < 0.01).

**Figure 2 cells-12-00721-f002:**
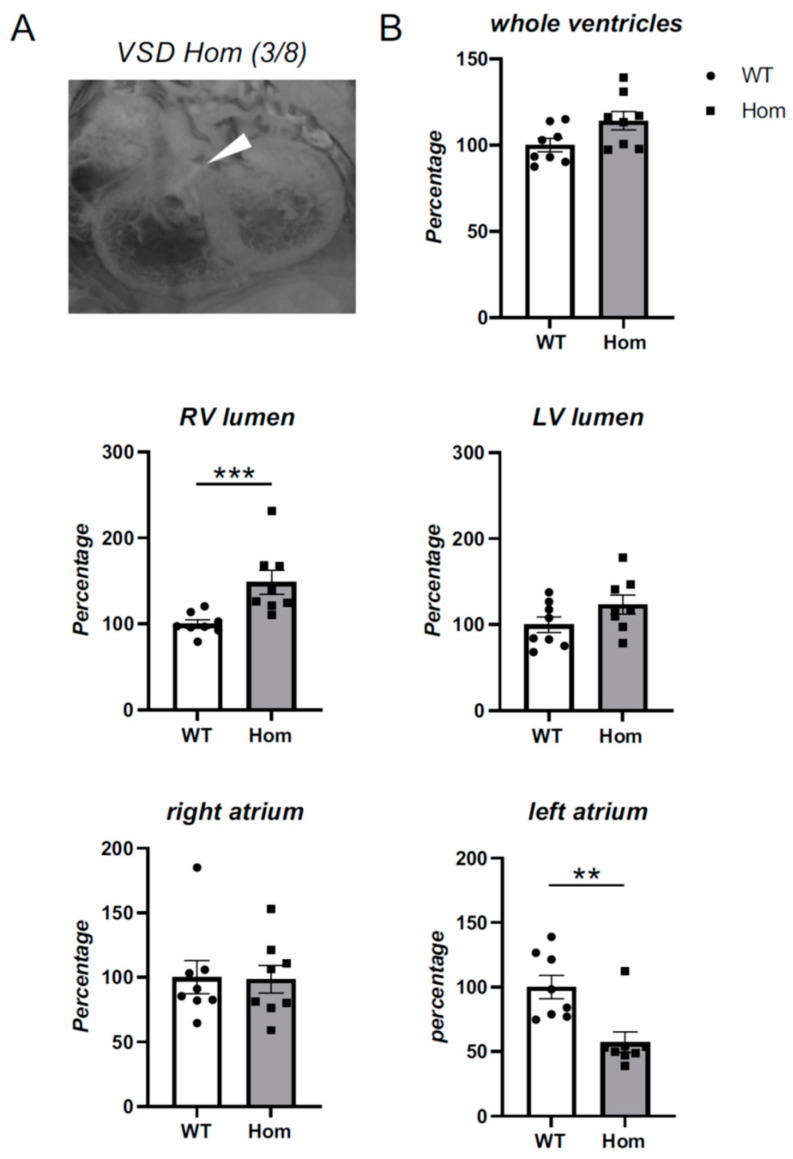
Homozygous embryos show gross structural heart defects. (**A**) HREM image of an E15.5 homozygous heart with a peri-membranous ventricular septal defect (VSD, white arrowhead). (**B**) Total volume of several heart compartments relative to control hearts was determined using Amira software. Percentage volume was assessed for whole ventricles (including myocardial wall and lumen), right (RV) and left ventricular (LV) lumens, right and left atria (including myocardial wall and lumen). Statistical significance was tested by one-tailed Fisher’s exact test (**A**), or Mann–Whitney tests. *n* = 8 WT (white bars), *n* = 8 Hom (grey bars) ** *p* < 0.01, *** *p* < 0.001.

**Figure 3 cells-12-00721-f003:**
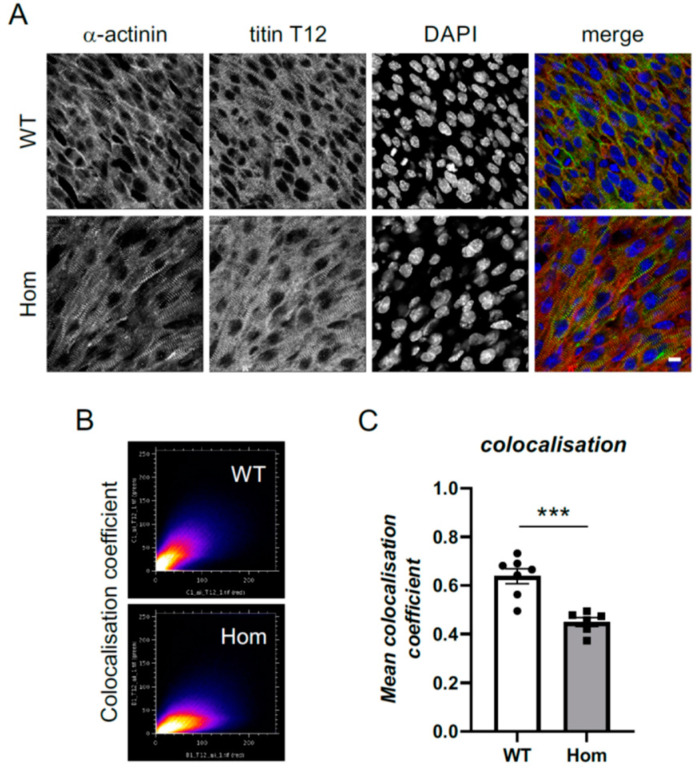
Localisation of alpha-actinin in sarcomeric structures in E15.5 hearts. (**A**) Ventricular wholemount staining for alpha-actinin (first column), titin Z-disk epitope T12 (second column), nuclei visualised by DAPI (third column). Last column shows merged images: alpha-actinin in green, titin T12 in red and DAPI in blue. Scale bar represents 10 microns. Both WT (first row) and Hom animals (second row) show sarcomeric alpha-actinin staining. (**B**) Colocalisation coefficient plots for alpha-actinin and titin T12 for the images in A (see Material and Methods). (**C**) Reduced colocalisation of alpha-actinin and titin T12 in Hom E15.5 hearts (nested ANOVA, *** *p* < 0.001). Each data point represents the average colocalisation coefficient for one heart, with a total of 43 images from 7 WT hearts and 25 images from 6 Hom hearts analysed.

**Figure 4 cells-12-00721-f004:**
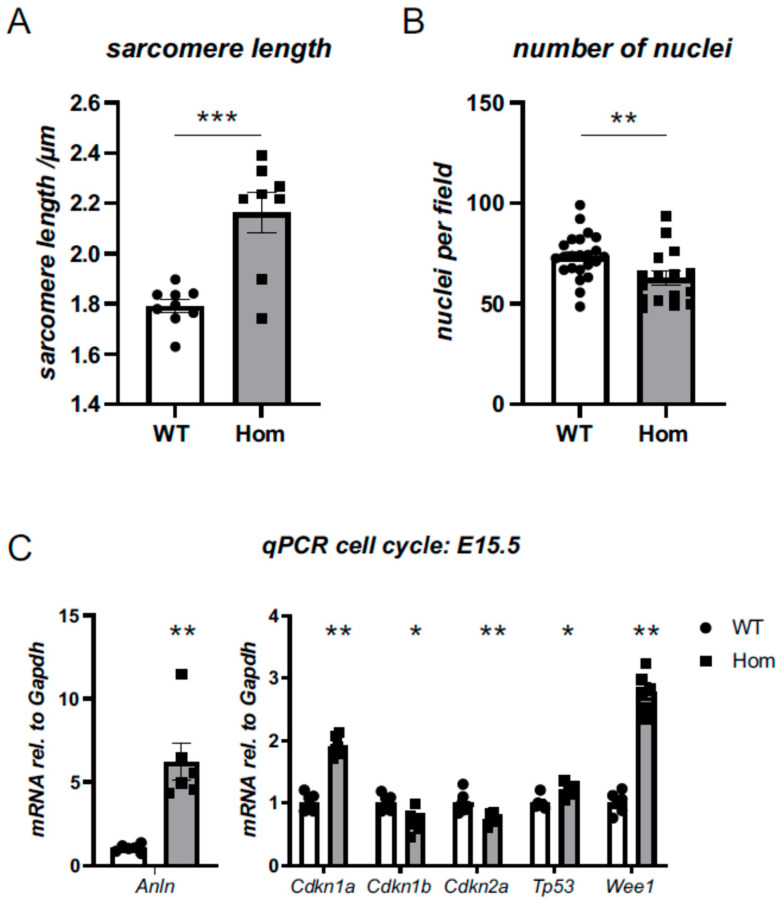
Measurement of cardiomyocyte sarcomere and nuclei parameters reveals alterations in Hom hearts. Wholemount stainings (see Figure 3A) for alpha-actinin, phalloidin (F-actin) and DAPI were analysed. (**A**) Hom hearts display increased mean sarcomere length. Each data point represents the average sarcomere length for one heart, with a total of 55 images from 9 WT hearts and 41 images from 8 Hom hearts analysed; nested ANOVA, *** *p* < 0.001. (**B**) Hom hearts have fewer nuclei. Each data point represents the average value for one heart, with a total of 112 images from 23 WT hearts and 79 images from 16 Hom hearts analysed; nested ANOVA, ** *p* < 0.01. For a wider range of nuclear parameters being analysed, see Appendix A. (**C**) Targeted assessment of transcriptional changes by qPCR in WT and Hom E15.5 for cell-cycle markers. All measurements are normalised to *Gapdh*. Significant changes are observed in the hearts of Hom mice for all transcripts investigated (Student’s *t*-test, *n* = 6 per group, * *p* < 0.05, ** *p* < 0.01).

**Figure 5 cells-12-00721-f005:**
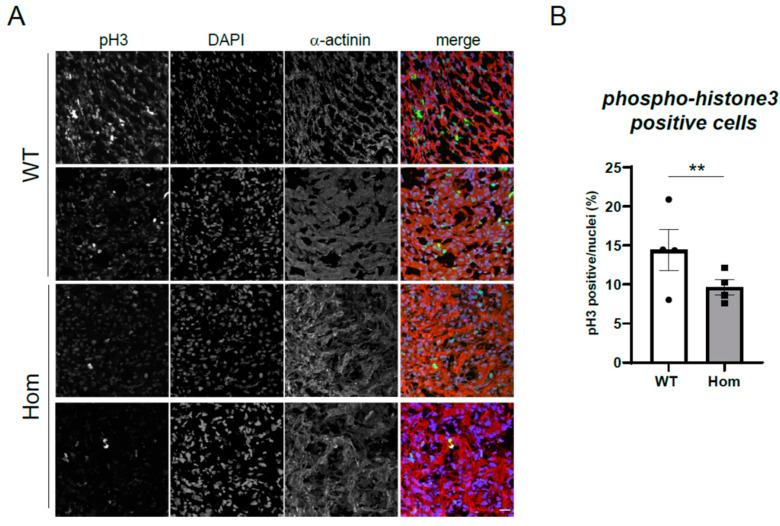
Cell cycling defects in E15.5 Hom hearts. (**A**) Cryosections of WT and Hom hearts were stained for cell proliferation marker phospho-histone H3 (pH3), sarcomeric alpha-actinin and nuclei were visualized with DAPI. Merged images show pH3 in green, DAPI in blue and alpha-actinin in red. Scale bar represents 20 microns. (**B**) Quantification of pH3 positive nuclei as a percentage of total nuclei (DAPI) is significantly reduced in E15.5 Hom hearts. ** *p* < 0.01 (nested ANOVA). Each data point represents the average value for one heart, with a total of 21 images from 4 WT hearts and 22 images from 4 Hom hearts analyzed.

**Figure 6 cells-12-00721-f006:**
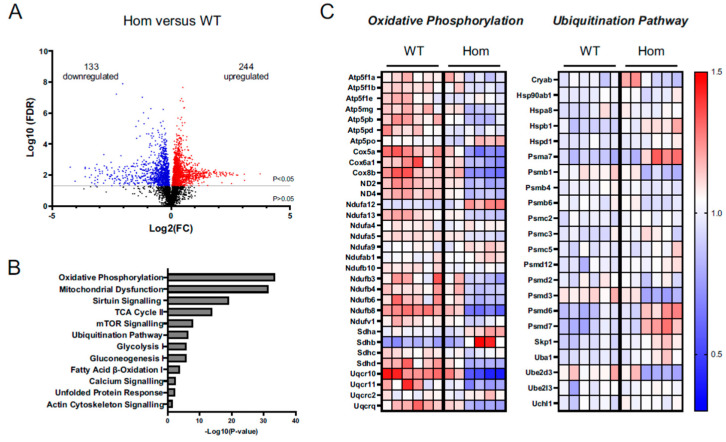
Mass spectroscopy (MS) analysis of WT and Hom hearts at E15.5 (*n* = 6 per group). (**A**) Alterations in protein levels between Hom and WT hearts are displayed in a volcano plot that shows 133 downregulated (blue) and 244 upregulated (red) using *p* < 0.05. Black dots represent protein fold changes at *p* > 0.05. (**B**) Selected hits of significantly dysregulated canonical pathways identified using IPA. (**C**) Heatmap of protein changes identified using IPA for oxidative phosphorylation and ubiquitination pathway. Colours range from red (increased) to blue (decreased) in WT and Hom hearts based on the relative abundance identified using MS.

**Figure 7 cells-12-00721-f007:**
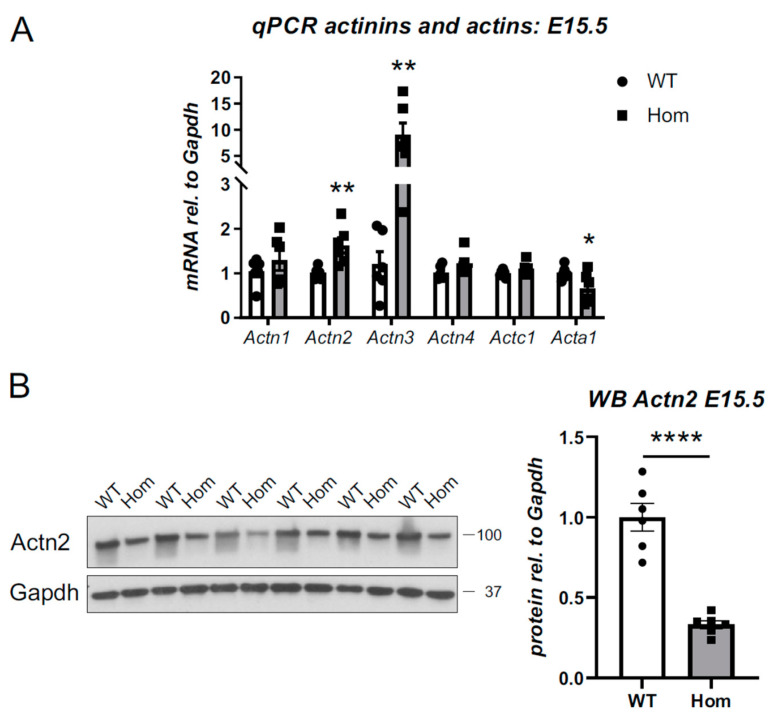
Actn2 destablisation at the protein level in Hom E15.5 hearts. (**A**) Targeted assessment of transcriptional changes by qPCR in WT and Hom E15.5 for actinins (Actn1-Actn4) and actins (cardiac actin Actc1 and skeletal muscle actin Acta1). All measurements are normalised to Gapdh. Significant upregulation of Actn2 and Actn3 as well as downregulation of Acta1 are observed in the hearts of Hom mice (Student’s *t*-test, *n* = 6 per group, * *p* < 0.05, ** *p* < 0.01)). (**B**) Reduced protein levels of alpha-actinin 2 in Hom E15.5 hearts. Left—Western blot for alpha-actinin 2 (Actn2) and Gapdh (*n* = 6 per group); position of marker bands indicated (molecular weight in kD). Right—Quantification indicates reduction of Actn2 protein levels in Hom E15.5 hearts (Student’s *t*-test, **** *p* < 0.0001).

## Data Availability

The mass spectrometry proteomics data have been deposited to the ProteomeXchange Consortium via the PRIDE partner repository with the dataset identifier PXD039226.

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
