# Peer review of "Insights into the Role of a Cardiomyopathy-Causing Genetic Variant in ACTN2"

_cells, 2023, doi:10.3390/cells12050721_

Round 1

Reviewer 1 Report

I have read the paper with great interest and have only two comments.

  1. At the beginning of the discussion (lines 438-443), the Authors refer to the variant segregation in the affected family. Based on the description, some readers might struggle to recognize the inheritance pattern. Please state that affected family members are heterozygotes or variant is inherited dominantly. Also, please comment on whether cases of homozygotes are known so far.
  2. Although it's not obligatory, I strongly suggest adjusting the sequence variants' nomenclature according to the HGVS recommendations.

Author Response

We thank the reviewer for their time and helpful comments.

Comments and Suggestions for Authors

I have read the paper with great interest and have only two comments.

  1. At the beginning of the discussion (lines 438-443), the Authors refer to the variant segregation in the affected family. Based on the description, some readers might struggle to recognize the inheritance pattern. Please state that affected family members are heterozygotes or variant is inherited dominantly. Also, please comment on whether cases of homozygotes are known so far.

This has now been clarified in the first two paragraphs of the discussion (L451-460).

  1. Although it's not obligatory, I strongly suggest adjusting the sequence variants' nomenclature according to the HGVS recommendations.

We have now applied the HGVS recommendation as suggested throughout the manuscript.

Reviewer 2 Report

This highly interesting and well written manuscript by Broadway-Stringer, Jiang et al. investigates the pathomechanism of HCM caused by a ACTN2 M228T mutation in a newly generated mouse model using state-of-the-art technology for morphological, functional and biochemical characterization of mouse hearts.

The authors found protein instability in homozygous mouse mutants due to enhanced UPS activity and suggest it as a driving factor in the pathogenesis. Further evidence would strengthen this conclusion, such as ubiquitination of alpha actinin (which was not detectable in whole heart lysates (Fig. S17), but maybe after immunoprecipitation of alpha actinin) and/or measurements of proteasomal proteolytic activity (caspase-like, trypsin-like, chymotrypsin-like).

Human HCM is caused by the heterozygous mutation. Is there any evidence for Actn2 destabilization also in heterozygous hearts (altered UPS activity, (ultra-)structural abnormalities etc.)? 

This argument is equally true for the finding of mitochondrial dysfunction. It should be considered that shortly before cardiac death myocardial energetic deficiency may also be a secondary event.

Minor:

There is some inconsistency about the phenotype (e.g., 3 out of 8 homozygous mice presented with VSD) that is probably attributable to the backcrossing of mice into a C57BL/6 genetic background. What is known about the (HCM) phenotype of ACTN2 M228T mutant mice in the original (inbred) background? At least heterozygous mice were generated.

Do ACTN2 M228T mutant mice show a skeletal muscle phenotype?

Author Response

We thank the reviewer for their time and helpful comments.

Comments and Suggestions for Authors

This highly interesting and well written manuscript by Broadway-Stringer, Jiang et al. investigates the pathomechanism of HCM caused by a ACTN2 M228T mutation in a newly generated mouse model using state-of-the-art technology for morphological, functional and biochemical characterization of mouse hearts.

The authors found protein instability in homozygous mouse mutants due to enhanced UPS activity and suggest it as a driving factor in the pathogenesis. Further evidence would strengthen this conclusion, such as ubiquitination of alpha actinin (which was not detectable in whole heart lysates (Fig. S17), but maybe after immunoprecipitation of alpha actinin) and/or measurements of proteasomal proteolytic activity (caspase-like, trypsin-like, chymotrypsin-like).

We have taken up the excellent advice and measured proteasomal proteolytic activities (new Fig. S20A). They are unchanged in the homozygous embryonic hearts and this is now briefly discussed in the manuscript (L533-536).

While this leaves a question mark on the role of UPS in the Actn2 mutant protein destablilisation, our proteomics data clearly demonstrate enrichment of UPS-related pathways (Fig. 6C). To highlight this we have added a new Table S8, looking specifically at UPS-related proteins in our proteomics dataset.

As for the suggested experiment of immuno-precipitating of alpha-actinin and probing for linked ubiquitin, we have employed similar experiments in the past, using tandem ubiquitin binding entities (TUBEs) [1, 2]. However, these experiments were previously performed from adult ventricular tissue (100 mg), which equates to 30 to 50 embryonic hearts. Moreover, they were only successful if a proteasomal inhibitor (MG-262 or epoxomicin) was applied in vivo before harvesting the tissue. By Home Office regulations, we are not allowed to carry out the same drug treatment in pregnant mice to treat the embryos.

We are confident that using a cellular model (induced pluripotent stem cell derived cardiomyocytes carrying the genetic variant) is the way forward, both ethically and practically: it will allow us to use inhibitors of proteolytic pathways to dissect the role of UPS and autophagy and generate sufficient sample material. While we plan to carry out this work in future, it is beyond the scope of the current manuscript. This is now briefly mentioned in the discussion (L534-536).

Human HCM is caused by the heterozygous mutation. Is there any evidence for Actn2 destabilization also in heterozygous hearts (altered UPS activity, (ultra-)structural abnormalities etc.)? 

We would like to point the reviewer at Fig. S4(A,B), which shows normal Actn2 protein levels in the heterozygous mice. In addition, we have now performed an ubiquitin blot (new Fig. S4C), which fails to identify any differences of ubiquitinylation between mature wildtype and Het mice.

Moreover, measurement of proteasomal activities showed no difference between hearts of these mice in a small pilot study (only n=2 per genotype available):

Fig. R1: Proteasomal activities measured in mature male mice (415 days old). Luminescent values are normalized to an average of 1 in the WT group. Only two hearts were available per genotype, no significant changes are observed (Student’s t-test).

No structural changes were observed on histology (Fig. S3), which is in agreement with a lack of an overt phenotype in the heterozygous mice.

We discuss the fact that heterozygous, late-onset human cardiomyopathy cannot always be reflected in mouse models in the manuscript (L462-465).

This argument is equally true for the finding of mitochondrial dysfunction. It should be considered that shortly before cardiac death myocardial energetic deficiency may also be a secondary event.

At this point we respectfully disagree. We believe the link of alpha-actinin integrity and mitochondrial function could be causative: an inducible, cell autonomous inactivation of alpha-actinin in individual cardiac cells leads to mitochondrial dysfuntion [3]. Mechanistically, alpha-actinin has been proposed to bring RNA transcripts coding for mitochondrial components to the proximity of the Z-disc via its binding to the RNA binding protein IGF2BP2.  In pathological conditions, such as cardiomyopathies, this interaction is required for proper mitochondrial function [4]. This is discussed in the manuscript (L562-564).

In addition, we have now added a comment of the possibility of myocardial energetic deficiency secondary to cell death [5] (L565-566).

Minor:

There is some inconsistency about the phenotype (e.g., 3 out of 8 homozygous mice presented with VSD) that is probably attributable to the backcrossing of mice into a C57BL/6 genetic background. What is known about the (HCM) phenotype of ACTN2 M228T mutant mice in the original (inbred) background? At least heterozygous mice were generated.

We would like to clarify the mice are congenic C57BL/6 (using C57BL/6N stem cells for the genome-editing). They were then backcrossed onto C57BL/6J for at least six generations (in case of homozygous animals) for phenotyping (simply because C57BL/6N breeders are not available in the UK anymore). This is clearly stated in the Methods section (L115-117).

The main difference between both C57BL/6 sub-strains is a natural mutation in the Nnt gene in some C57BL/6J lineages [6]. We have checked for this Nnt mutation and it is consistently present in all mice of this colony (Suppl methods p. 2). Hence, it cannot explain the inconsistency among the homozygous mice.

It is common for genetic mouse models of congenital heart disease to have variable penetrance of phenotypes in a completely congenic background, see e.g. [7]. This may reflect the stochastic nature of some of the events in embryonic heart development.

Do ACTN2 M228T mutant mice show a skeletal muscle phenotype?

There is no evidence for a skeletal muscle phenotype: mice have normal mobility, behaviour, and body weights. We have now performed microscopic analysis (new Fig. S5) of skeletal muscle, which shows normal fibre sizes and absence of centrally located nuclei (the latter would be a hallmark of skeletal muscle pathologies). The lack of a skeletal muscle phenotype is in agreement with the absence of skeletal muscle involvement in the original family of patients.

References Reviewer 2:

  1. Ehsan, M., et al., Mutant Muscle LIM Protein C58G causes cardiomyopathy through protein depletion. J Mol Cell Cardiol, 2018. 121: p. 287-296.
  2. Jiang, H., et al., Functional analysis of a gene-edited mouse model to gain insights into the disease mechanisms of a titin missense variant. Basic Res Cardiol, 2021. 116(1): p. 14.
  3. Guo, Y., et al., Sarcomeres regulate murine cardiomyocyte maturation through MRTF-SRF signaling. Proc Natl Acad Sci U S A, 2021. 118(2).
  4. Ladha, F.A., et al., Actinin BioID reveals sarcomere crosstalk with oxidative metabolism through interactions with IGF2BP2. Cell Rep, 2021. 36(6): p. 109512.
  5. Orogo, A.M. and A.B. Gustafsson, Cell death in the myocardium: my heart won't go on. IUBMB Life, 2013. 65(8): p. 651-6.
  6. Freeman, H.C., et al., Deletion of nicotinamide nucleotide transhydrogenase: a new quantitive trait locus accounting for glucose intolerance in C57BL/6J mice. Diabetes, 2006. 55(7): p. 2153-6.
  7. Wilson, R., et al., Highly variable penetrance of abnormal phenotypes in embryonic lethal knockout mice. Wellcome Open Res, 2016. 1: p. 1.

Reviewer 3 Report

In this study, Broadway-Stringer et al. used Crisp-Cas9 technology and generated a mouse line of Actn2 M228T. They conducted a detailed analysis on the hetero and homo Actn2 M228T embryo/adult mice. They found hetero mice have almost no defect in heart function, except that male hetero mice (at 38 weeks) have increased expressions of several heart malfunction related genes, such as Fhl1. Furthermore, they went on to study the mechanism on heart failure of hom at E15.5. Through qPCR and MS, they found the cardiomyocytes from hom mice have cell cycle defect and mitochondria and ubiquitination system may be responsible for the heart failure. Generally, the study is interesting and gave some insights in human heart pathology with Actn2 mutation.

The following points should be discussed, clarified or experimentally investigated:

1.     Fig 3A, nucleus of hom heart seems bigger than those in WT group. Any explanation? How this phenotype contributes to heart failure?

2.     Fig 4C: WB analysis on these cell cycle regulators is highly recommended; IF analysis on cardiomyocyte proliferation, cell death should be conducted.

3.     Fig 5: The author used proteomic approach to identify the heart defect mechanism and found that oxidative phosphorylation in mitochondria and ubiquitination pathways may be responsible. It seems to the reviewer that reduction in oxidative phosphorylation related proteins is convincing, while an active UB system is not evident as shown in the heatmap. Limited numbers of Ub proteins showed significant and consistent changes between WT and hom samples.

4.     In hom heart, mRNA of Actn2 M228T is high, while protein of Actn2 M228T is low. The authors researched a hypothesis that proteasome degradation is responsible. However, only a WB of WT Actn2 and Actn2 M228T is shown. In order to test the hypothesis above, cell culture with proteosome inhibitor should be used. Furthermore, in Fig6A, qPCR indicates that a significant changes of actn2 transcripts in hom samples, and it is quite mild (around 1.2-1.3 fold in average). It would be ideal to use several pairs of qPCRs to quantify mRNA level of actn2.

Author Response

We thank the reviewer for their time and helpful comments.

Comments and Suggestions for Authors

In this study, Broadway-Stringer et al. used Crisp-Cas9 technology and generated a mouse line of Actn2 M228T. They conducted a detailed analysis on the hetero and homo Actn2 M228T embryo/adult mice. They found hetero mice have almost no defect in heart function, except that male hetero mice (at 38 weeks) have increased expressions of several heart malfunction related genes, such as Fhl1. Furthermore, they went on to study the mechanism on heart failure of hom at E15.5. Through qPCR and MS, they found the cardiomyocytes from hom mice have cell cycle defect and mitochondria and ubiquitination system may be responsible for the heart failure. Generally, the study is interesting and gave some insights in human heart pathology with Actn2 mutation.

The following points should be discussed, clarified or experimentally investigated:

Fig 3A, nucleus of hom heart seems bigger than those in WT group. Any explanation? How this phenotype contributes to heart failure?

While we agree nuclei look larger in this particular image (which was selected for the abnormal dividing cell), the overall nuclear size is unchanged in the Hom embryonic hearts (see Table S5).

We have replaced this figure with a new one (now Fig. S17).

Fig 4C: WB analysis on these cell cycle regulators is highly recommended; IF analysis on cardiomyocyte proliferation, cell death should be conducted.

We thank the reviewer for the excellent suggestion and have carried out the following:

We have performed Western blotting for p53 (Tp53), now shown in Fig. S16. It shows a trend towards increased p53 levels in the Hom embryonic hearts, but does not reach significance. We have also attempted to blot for p21 (Cdkn1a), but unfortunately could not get the antibody to work.

We have also performed staining for phospho-histone3 as a marker for cell proliferation and observe a significant reduction in cells positive for this marker in homozygous embryonic hearts (new Fig. 5).

Higher magnification of proliferating cells (new Fig. S17) shows that the E15.5 homozygous cardiomyocytes that are dividing as indicated by being positive for phosphorylated histone and the clear evidence of metaphase chromosome arrangement, but they appear to have a defect in myofibril disassembly (bottom half of figure), which is complete at this state in the wildtype cardiomyocytes (top half of figure). These results are reminiscent of results shown in [1] of cardiomyocytes treated with the proteasome inhibitor MG132 (Figure 7) and in [2] in cardiomyocytes treated with calpain inhibitor (Figure 8), suggesting that the impaired myofibrillar breakdown might arrest the progress of cytokinesis.

Further, we have performed TUNEL stain as a marker of apoptosis: TUNEL positive cells were negligible independent of genotype, while the positive control (DNase treated) worked well. We conclude apoptosis does not play a major role in the model and have decided not to show the TUNEL stain.

As part of measuring proteolytic activities, we have also measured caspase-like activity (Fig. S20A), which is linked to apoptosis, but unchanged in the in the Hom embryonic hearts.

Fig 5: The author used proteomic approach to identify the heart defect mechanism and found that oxidative phosphorylation in mitochondria and ubiquitination pathways may be responsible. It seems to the reviewer that reduction in oxidative phosphorylation related proteins is convincing, while an active UB system is not evident as shown in the heatmap. Limited numbers of Ub proteins showed significant and consistent changes between WT and hom samples.

We agree with the reviewer that the data on the role of the UPS are not as clear as we would like them to be. However, the proteomics dataset clearly shows enrichment of UPS-related proteins. This is now highlighted in a new Table S8.

We have further measured proteolytic activities in the embryonic hearts (Fig. S20A). However, they are unchanged and this is now briefly discussed in the manuscript (L 533-536).

In hom heart, mRNA of Actn2 M228T is high, while protein of Actn2 M228T is low. The authors researched a hypothesis that proteasome degradation is responsible. However, only a WB of WT Actn2 and Actn2 M228T is shown. In order to test the hypothesis above, cell culture with proteosome inhibitor should be used.

We agree with the reviewer that the use of proteasomal inhibitors is an excellent idea and will help to identify the protein degrading pathways responsible.

Previously, we have applied proteasomal inhibitors (MG-262 or epoxomicin) in vivo in mouse models with protein destabilisation [3, 4]. However, by Home Office regulations, we are not allowed to carry out such drug treatments in pregnant mice to treat the embryos.

It is challenging to generate primary cardiomyocyte cultures from E15.5 embryos, due to the tiny size of the heart at this time point (2-3 mg). Further, we do not know the genotype of the embryos until a week later, hence inhibitor/control pairs within the same litter are not possible.

Hence we are planning to generate an induced pluripotent stem cell derived cardiomyocyte line with this genetic variant for further mechanistic studies (including the use of proteasomal and autophagy inhibitors), but this is beyond the scope of the current manuscript. The use of such inhibitors is now briefly discussed in the manuscript (L534-536).

Furthermore, in Fig6A, qPCR indicates that a significant changes of actn2 transcripts in hom samples, and it is quite mild (around 1.2-1.3 fold in average). It would be ideal to use several pairs of qPCRs to quantify mRNA level of actn2.

We have now performed qPCR with two further Actn2 TaqMan probes (new Fig. S19A). They show a similar extent of upregulation as the original probe used in Fig. 7A.

       Probe           Assay ID                    Coverage       Hom upregulation (mean +/- SEM)

          #1               Mm00473657_m1      exons 11-12                        163 +/- 18 %

          #2               Mm01340071_m1      exons 16-17                        141 +/- 8 %

          #3               Mm01340076_mH     exons 3-4                            132 +/- 14 %

This mild increase of the transcript, discussed as a compensatory mechanisms (L523-525), is in agreement with a similar extent seen for Csrp3 p.Cys58Gly [3], shown as mean +/- SEM):

Csrp3 p.Cys58Gly mouse                Transcript                              Protein          

Heterozygous                                    up to 114 +/- 4 %                   down to 54 +/- 5 %

Homozygous                                     up to 253 +/- 21 %                 down to 15.2 +/- 1.4 %

There are parallels between the Actn2 p.Met228Thr and Csrp3 p.Cys58Gly models: Both were described as autosomal dominant (heterozygous) variants in late-onset HCM patients, but have no overt phenotype in mice in the heterozygous setting. Further, both have a striking phenotype in the homozygous mouse model, with protein destabilization occurring accompanied by mild upregulation of the transcript.

References reviewer 3

  1. Ahuja, P., et al., Sequential myofibrillar breakdown accompanies mitotic division of mammalian cardiomyocytes. J Cell Sci, 2004. 117(Pt 15): p. 3295-306.
  2. Ahuja, P., et al., Re-expression of proteins involved in cytokinesis during cardiac hypertrophy. Exp Cell Res, 2007. 313(6): p. 1270-83.
  3. Ehsan, M., et al., Mutant Muscle LIM Protein C58G causes cardiomyopathy through protein depletion. J Mol Cell Cardiol, 2018. 121: p. 287-296.
  4. Jiang, H., et al., Functional analysis of a gene-edited mouse model to gain insights into the disease mechanisms of a titin missense variant. Basic Res Cardiol, 2021. 116(1): p. 14.

Round 2

Reviewer 2 Report

The authors adequately addressed all of my questions. I have no further comments.

Reviewer 3 Report

The authors did excellent revision job. Now the manuscript should be accepted for publication!